# Post-Translational Modification Analysis of VDAC1 in ALS-SOD1 Model Cells Reveals Specific Asparagine and Glutamine Deamidation

**DOI:** 10.3390/antiox9121218

**Published:** 2020-12-02

**Authors:** Maria Gaetana Giovanna Pittalà, Simona Reina, Salvatore Antonio Maria Cubisino, Annamaria Cucina, Beatrice Formicola, Vincenzo Cunsolo, Salvatore Foti, Rosaria Saletti, Angela Messina

**Affiliations:** 1Department of Biological, Geological and Environmental Sciences, Molecular Biology Laboratory, University of Catania, Via S. Sofia 64, 95123 Catania, Italy; marinella.pitt@virgilio.it (M.G.G.P.); simona.reina@unict.it (S.R.); salvatore.cubisino@phd.unict.it (S.A.M.C.); 2we.MitoBiotech.srl, c.so Italia 172, 95129 Catania, Italy; 3Department of Chemical Sciences, Organic Mass Spectrometry Laboratory, University of Catania, Via S. Sofia 64, 95123 Catania, Italy; annamaria.cucina@phd.unict.it (A.C.); vcunsolo@unict.it (V.C.); sfoti@unict.it (S.F.); 4School of Medicine & Surgery, Nanomedicine Center NANOMIB, University of Milano-Bicocca, 20900 Monza, Italy; b.formicola@campus.unimib.it

**Keywords:** deamidation, amyotrophic lateral sclerosis, voltage dependent anion channel, post-translational modifications, mitochondria, ROS, mass spectrometry analysis, Orbitrap fusion tribrid, neurodegeneration, SOD1

## Abstract

Mitochondria from affected tissues of amyotrophic lateral sclerosis (ALS) patients show morphological and biochemical abnormalities. Mitochondrial dysfunction causes oxidative damage and the accumulation of ROS, and represents one of the major triggers of selective death of motor neurons in ALS. We aimed to assess whether oxidative stress in ALS induces post-translational modifications (PTMs) in VDAC1, the main protein of the outer mitochondrial membrane and known to interact with SOD1 mutants related to ALS. In this work, specific PTMs of the VDAC1 protein purified by hydroxyapatite from mitochondria of a NSC34 cell line expressing human SOD1G93A, a suitable ALS motor neuron model, were analyzed by tryptic and chymotryptic proteolysis and UHPLC/High-Resolution ESI-MS/MS. We found selective deamidations of asparagine and glutamine of VDAC1 in ALS-related NSC34-SOD1G93A cells but not in NSC34-SOD1WT or NSC34 cells. In addition, we identified differences in the over-oxidation of methionine and cysteines between VDAC1 purified from ALS model or non-ALS NSC34 cells. The specific range of PTMs identified exclusively in VDAC1 from NSC34-SOD1G93A cells but not from NSC34 control lines, suggests the appearance of important changes to the structure of the VDAC1 channel and therefore to the bioenergetics metabolism of ALS motor neurons. Data are available via ProteomeXchange with identifier <PXD022598>.

## 1. Introduction

Amyotrophic lateral sclerosis (ALS) is a devastating neurodegenerative disease caused by progressive degeneration of the motor neurons in the brainstem and spinal cord that leads patients to death by respiratory paralysis within 2–5 years of onset [1]. While about 10% of ALS cases are associated with genetic defects, 90% of ALS cases are sporadic. Several genetic risk factors are also implicated in sporadic ALS (sALS) [2].

Oxidative stress-induced damage is a major mechanism in the ALS pathophysiology and can result from an imbalance between free radical production and degradation. Free radicals, such as reactive oxygen species (ROS) and reactive nitrogen species (RNS), are generated from various cell sources. Mitochondria are the main fount of ROS, while a smaller proportion of ROS and RNS is produced by non-mitochondrial oxidative enzymes [3]. Interestingly, non-mitochondrial ROS usually function as signaling molecules and are seldom involved in pathophysiological processes [4]. However, ROS produced through Nox enzymes have been implicated in ALS pathogenesis [5]. Free radicals cause oxidative damage to lipids, proteins, and nucleic acids contributing thus to trigger or amplify the pathological mechanisms related to ALS [6]. Therefore, mitochondrial dysfunction is both the main contributor to oxidative stress and its main consequence.

The Cu/Zn superoxide dismutase (SOD1) associates with about 20% of familial ALS (fALS) cases and over 180 mutant forms of enzymatically active or inactive SOD1 have been characterized in humans (http://alsod.iop.kcl.ac.uk) [7]. In affected tissues, toxic effects of SOD1 mutants are related to the formation of misfolded SOD1 aggregates upon the mitochondrial surface, leading to morphological degeneration and malfunctioning of the organelle [8,9].

In the spinal cord from ALS patients, voltage dependent anion selective channel isoform 1 (VDAC1) represents the docking site on the outer mitochondrial membrane for ALS-linked SOD1 mutants [10,11].

VDAC1 is the most abundant protein of the outer mitochondrial membrane and is evolutionary preserved from yeast to man [12,13,14]. As a voltage-dependent anionic channel, it carries ATP, ions, and other small metabolites. It is responsible for metabolic exchanges to and from the mitochondrion, thus controlling cell metabolism and mitochondrial function. VDAC1 also acts as a scaffold for important molecules, such as hexokinase and proteins from the Bcl2 family which regulate metabolism and apoptosis. Apoptotic stimuli also induce VDAC1 oligomerization that leads to mitochondrial permeability and apoptosis [15]. Therefore, due to its important role, dysfunction of the VDAC channel can lead to various diseases, such as cancer and neurodegenerative diseases like Parkinson’s disease, Alzheimer’s disease, and ALS [16,17].

In ALS, the VDAC1-SOD1 mutant interaction strongly affects the functional properties of VDAC1 channel suggesting a role in the impairment of the bioenergetics metabolism and oxidative stress of ALS motor neurons [10]. It is also known that low levels of hexokinase I (HK1) in the spinal cord make motor neurons more susceptible to ALS in comparison to other tissues. In particular, this reduction of HK1 levels increases the availability of VDAC1 to interact with mutant SOD1, thereby facilitating mitochondrial dysfunction and cell death [10].

Protein-protein interactions (PPI) at the surface of mitochondria are an emerging novelty nowadays, but still few information is available. The mitochondria-associated proteins interact with the organelle orchestrating its response in order to maintain proper cellular function. In fact, alterations in selective PPIs at the mitochondrial surface are the result of pathological associations that drive the progression of neurodegenerative disorders [18]. Protein PTMs guide the cellular response to specific insults, distinguishing toxic PTMs (e.g., oxidations) and protective PTMs (e.g., cysteinylation); high levels of toxic changes or a varied set of PTMs are related to disease conditions.

Under physiological conditions, VDAC1 presents several post-translational changes (phosphorylation [19], acetylation [20], tyrosine nitration [21] and oxidation of cysteine and methionine) whose role in protein activity has only been partially clarified [22,23,24]. Moreover, VDAC1, like other mitochondrial proteins in the outer membrane, is a target for Parkin-mediated ubiquitination [25]. Ubiquitination, like phosphorylation and acetylation, is a reversible modification that allows the selection of dysfunctional mitochondria for clearance and regulates quality control pathways.

Recent advances in the study of molecular markers of ageing, hypoxia, and age-related neurodegenerative diseases have highlighted the important role of protein deamidation. As of today, no one studied possible deamidation of the VDAC channel.

The abnormal accumulation of misfolded proteins and dysfunctional mitochondria is a distinctive feature of ALS and many neurodegenerative diseases [26]. In addition, mitochondrial proteins undergo further PTMs in response to both physiological and pathological cell signals. It is known that PTMs influence the activity of the VDAC channel and thus the mitochondrial function and its fate [19,27]. For example, cysteines undergo reversible and irreversible redox reactions by ROS and are important for the stabilization of the VDAC structure [22,28].

In this work, by using a high-resolution mass spectrometry analysis we identified non-reversible post-translational changes in VDAC1 that may be involved in its specific interaction with ALS-related SOD1 mutants. Our findings allow us to hypothesize its further role in the disease and as a marker for irreparably damaged mitochondria.

## 2. Materials and Methods

### 2.1. Chemicals

All chemicals were of the highest purity commercially available and were used without further purification. Ammonium bicarbonate, calcium chloride, phosphate-buffered saline (PBS), Tris-HCl, Triton X-100, sucrose, mannitol, ethylene glycol tetraacetic acid (EGTA), ethylenediaminetetraacetic acid (EDTA), formic acid (FA), dithiothreitol (DTT) and iodoacetamide (IAA) were obtained from Sigma-Aldrich (Milan, Italy). High-glucose DMEM (Dulbecco’s Modified Eagle Medium) and fetal bovine serum (FBS) were obtained from Gibco-Thermo Fisher Scientific (Milan, Italy). DMEM F12 and tetracycline-free FBS were obtained from Euro Clone. G418 and Doxycycline were obtained from Carlo Erba and Sigma-Aldrich (Milan, Italy), respectively. Trypsin/EDTA (for cell cultures) and penicillin-streptomycin (P/S) were purchased from Invitrogen. All the other stock solutions for cell culturewere from Euroclone (Milan, Italy). Modified porcine trypsin and chymotrypsin were purchased from Promega (Milan, Italy). Water and acetonitrile (OPTIMA^®^ LC/MS grade) for LC/MS analyses were provided from Fisher Scientific (Milan, Italy).

### 2.2. NSC34 Cell Lines

The mouse motor neuron-like NSC34 cell line were from CELLutions Biosystem Inc., Burlington, ON, Canada) and NSC34 cells stably transfected with pTet-ON plasmid (Clontech, Mountain View, CA, USA) harboring sequences encoding SOD1 wt (NSC34-SOD1WT) or G93A (NSC34-SOD1G93A) were used as non-ALS motor neuron line and ALS motor neuron line, respectively [9]. Cell maintenance, induction, and transfection condition were as in Magrì et al., 2016 [10].

### 2.3. Extraction of Mitochondrial Proteins from NSC34 Cells under Reducing Condition

NSC34, NSC34-SOD1WT and NSC34-SOD1G93A cell lines were cultured in monolayer (75 cm^2^ tissue culture flask) until 75% confluence. For mitochondria preparation about 80 million cells/culture were used; NSC34 cells were harvested by trypsin with 0.25% EDTA and washed with PBS three times before disruption. Every PBS wash was eliminated by centrifugation at 1500× *g* for 5 min at 4 °C. Purification of mitochondria from NSC34-SOD1WT and NSC34-SOD1G93A cell lines was carried out 48 h after doxycycline induction of human SOD1 protein expression.

The total cell pellet obtained was resuspended in 1 mL of hypotonic buffer (200 mM mannitol, 70 mM sucrose, 10 mM HEPES pH 7.5, 1 mM EGTA pH 8.0). The cells were incubated in ice for 10 min and then lysed. The lysate obtained was centrifuged (700× *g* for 25 min at 4 °C) to eliminate the non-lysed cells and the nuclei. To increase the yield, after recovering the supernatant, the pellet containing the mitochondria was suspended in hypotonic buffer. Again, the suspension was first lysed and then centrifuged (700× *g* for 25 min at 4 °C).

The resulting supernatants from the two centrifugations were combined and centrifuged at higher speed (7000× *g* for 15 min at 4 °C). The supernatant, containing the cytoplasmic fraction, was removed while the pellet was washed with hypotonic buffer.

The suspension was then centrifuged at high speed (7000× *g* for 15 min at 4 °C) and at the end, after removing the supernatant, the pellet containing the mitochondria, was resuspended in 500 μL of hypotonic buffer and stored at 4 °C.

The total yield (in 500 μL of hypotonic buffer) was determined by Bicinchoninic Acid Protein Assay (BCA method) resulting in 0.474 mg, 0.524 mg, 0.628 mg for NSC34, NSC34-SOD1WT, and NSC34-SOD1G93A, respectively. The hypotonic buffer was then removed by centrifugation (10,000× *g* for 20 min at 4 °C). Reduction/alkylation was carried on before VDACs purification to avoid any possible artifact due to air exposure. 0.237 mg, 0.262 mg and 0.314 mg of protein from intact mitochondria purified from NSC34, NSC34-SOD1WT and NSC34-SOD1G93A, respectively, were incubated for 3 h at 4 °C in 1 mL of Tris-HCl 10 mM (pH 8.3) containing 0.00474 mmol, 0.00524 mmol and 0.00628 mmol of DTT for each cell line: this corresponds to an excess of 50:1 (mol/mol) over the estimated protein thiol groups. The temperature was kept at 4 °C to avoid possible reduction of methionine sulfoxide to methionine by methionine sulfoxide reductase. The alkylation was performed by the addition of IAA at the 2:1 M ratio over DTT for 1 h in the dark at 25 °C. Mixture was centrifuged for 30 min at 10,000× *g* at 4 °C and the pellet was stored at −80 °C until further use.

Reduced and alkylated intact mitochondria were lysed in buffer A (10 mM TrisHCl, 1 mM EDTA, 3% Triton X-100, pH 7.0) in ratio 5:1 (mitochondria mg/buffer volume mL) [29] for 30 min on ice and centrifuged at 17,400× *g* for 30 min at 4 °C. The supernatant containing mitochondrial proteins was subsequently loaded onto a homemade glass column 5 × 80 mm, packed with 0.6 g of dry hydroxyapatite (Bio-Gel HTP, Biorad). The column was eluted with buffer A at 4 °C and fractions of 500 μL were collected and tested for protein content by a fluorometer assay (Invitrogen Qubit™ Protein Assay kit, ThermoFisher Scientific, Milan, Italy). Fractions containing proteins were pooled and dried under vacuum. The hydroxyapatite eluate was divided into two aliquots, which were reduced to less than 100 µL and purified from non-protein contaminating molecules with the PlusOne 2-D Clean-Up kit (GE Healthcare Life Sciences, Milan, Italy) according to the manufacturer’s instructions. The desalted protein pellet was suspended in 100 μL of 50 mM ammonium bicarbonate (pH 8.3) and incubated at 4 °C for 15 min. Next, 100 μL of 0.2% RapiGest SF (Waters, Milan, Italy) in 50 mM ammonium bicarbonate were added and the samples were incubated at 4 °C for 30 min. Another aliquot of desalted protein pellet was suspended in 100 μL of 100 mM Tris-HCl, 10mM calcium chloride (pH 8.0) and incubated at 4 °C for 15 min. Next, 100 μL of 0.2% RapiGest SF in 100 mM Tris-HCl and 10 mM calcium chloride were added, and the samples were kept at 4 °C for 30 min. For each fraction, the recovered protein amount was determined in 100 μg by using a fluorometer assay. Reduced and alkylated proteins were then subjected separately to digestion using modified porcine trypsin and chymotrypsin, respectively. Tryptic digestion was carried out at an enzyme-substrate ratio of 1:50 at 37 °C for 4 h. Chymotryptic digestion was performed in Tris-HCl 100 mM, 10 mM calcium chloride (pH 8.0) at an enzyme-substrate ratio of 1:20, overnight at 25 °C.

### 2.4. Liquid Chromatography and Tandem Mass Spectrometry (LC–MS/MS) Analysis

Mass spectrometry data were acquired in triplicate for each sample assayed on an Orbitrap Fusion Tribrid (Q-OT-qIT) mass spectrometer (ThermoFisher Scientific, Bremen, Germany) equipped with a ThermoFisher Scientific Dionex UltiMate 3000 RSLCnano system (Sunnyvale, CA, USA), to assess the reproducibility of the available MS data. Samples obtained by in-solution tryptic and chymotryptic digestion were reconstituted in 30 μL of 5% FA aqueous solution and 1 μL was loaded onto an Acclaim^®^Nano Trap C18 column (100 μm i.d. × 2 cm, 5 μm particle size, 100 Å). After washing the trapping column with solvent A (H_2_O + 0.1% FA) for 3 min at a flow rate of 7 μL/min, the peptides were eluted from the trapping column onto a PepMap^®^ RSLC C18 EASY Spray, 75 μm × 50 cm, 2 μm, 100 Å column and were separated by elution at a flow rate of 0.250 μL/min, at 40 °C, with a linear gradient of solvent B (CH_3_CN + 0.1% FA) in A from 5% to 20% in 32 min, followed by 20% to 40% in 30 min, and 40% to 60% in 20 min. Eluted peptides were ionized by a nanospray (Easy-spray ion source (ThermoFisher Scientific, Bremen, Germany), using a spray voltage of 1.7 kV and introduced into the mass spectrometer through a heated ion transfer tube (275 °C). Survey scans of peptide precursors in the *m/z* range 400–1600 were performed at a resolution of 120,000 (@200 *m/z*) with an AGC target for the Orbitrap survey of 4.0 × 10^5^ and a maximum injection time of 50 ms. Tandem MS was performed by isolation at 1.6 Th with the quadrupole, and high energy collisional dissociation (HCD) was performed in the Ion Routing Multipole (IRM), using a normalized collision energy of 35 and rapid scan MS analysis in the ion trap. Only those precursors with charge state 1–3 and intensity above the threshold of 5 × 10^3^ were sampled for MS^2^. The dynamic exclusion duration was set to 60 s with a 10-ppm tolerance around the selected precursor and its isotopes. Monoisotopic precursor selection was turned on. AGC target and maximum injection time for MS/MS spectra were 1.0 × 10^4^ and 100 ms, respectively. The instrument was run in top speed mode with 3 s cycles, meaning the instrument would continuously perform MS^2^ events until the list of non-excluded precursors diminishes to zero or 3 s, whichever is shorter. MS/MS spectral quality was enhanced enabling the parallelizable time option (i.e., by using all parallelizable time during full scan detection for MS/MS precursor injection and detection). Mass spectrometer calibration was performed using the Pierce^®^ LTQ Velos ESI Positive Ion Calibration Solution (Thermo Fisher Scientific, Milan, Italy). MS data acquisition was performed using the Xcalibur v. 3.0.63 software (Thermo Fisher Scientific, Milan, Italy).

### 2.5. Database Search

LC–MS/MS data were processed by PEAKS de novo sequencing software (v. X, Bioinformatics Solutions Inc., Waterloo, ON, Canada). The data were searched against the 17,463 entries “Mus musculus” SwissProt database (release November 2019). The common Repository of Adventitious Proteins (c-RAP) contaminant database was included in the database search. Full tryptic or chymotrypsin peptides with a maximum of 3 missed cleavage sites were subjected to a bioinformatic search. Cysteine carboxyamidomethylation was set as the fixed modification, whereas oxidation of methionine, trioxidation and succination of cysteine, phosphorylation of serine, threonine and tyrosine, ubiquitin (*m/z* 114.0429) and ubiquitination (*m/z* 383.2281) of lysine, transformation of *N*-terminal glutamine and *N*-terminal glutamic acid residues in the form of pyroglutamic acid and *N*-terminal protein acetylation were included as variable modifications. The precursor mass tolerance threshold was 10 ppm, and the maximum fragment mass error was set to 0.6 Da. Peptide spectral matches (PSM) were validated using Target Decoy PSM Validator node based on q-values at a 0.1% False Discovery Rate (FDR). A protein was considered identified with a minimum of two peptides of which at least one had to be “unique”. Proteins containing the same peptides which could not be differentiated based on MS/MS analysis alone were grouped to satisfy the principles of parsimony. Finally, all the protein hits obtained were processed by using the InChorus function of PEAKS. This tool combines the database search results of PEAKS software with those obtained by the Mascot search engine with the aim not only to increase the coverage but also the confidence since the engines use independent algorithms and therefore the results confirm each other.

The raw data were also processed by MaxQuant 1.6.3.4 to further investigate deamidation. To this end, MaxQuant gives information about intensity of peptides in both unmodified and modified forms, allowing for an estimation of the deamidation ratio. The main parameters of the previous search were maintained: database, contaminant database, type of digestion, fixed and variable modifications. The principles of parsimony were still applied. Match type was “match from and to”. The decoy mode was “revert”. PSM, Protein and Site decoy fraction FDR were set at 0.01 as it was the threshold for peptide and protein identifications. All the other parameters were set as default.

#### 2.5.1. Identification of Deamidation Sites on VDAC1

An estimation of the percentage of deamidation in VDAC1 for each cell line was calculated with the aid of a freely available command-line script for Python 2.x (https://github.com/dblyon/deamidation), which uses the MaxQuant “evidence.txt” file. It has to be considered that the method is intended for comparative purposes and not as an absolute measure of deamidation.

MaxQuant’s “evidence.txt” file was used to calculate separate deamidation rates for Asparagine (N) and Glutamine (Q). The fraction of num_N (number of Asparagines) to num_N-2-D (number of deamidated Asparagines) and the fraction num_Q (number of Glutamines) to num_Q-2-E (number of deamidated Glutamines) were calculated for each peptide-to-spectrum match (PSM). The values obtained were termed respectively ratio_N-2-D and ratio_Q-2-E. The ratio_N-2-D or ratio_Q-2-E was multiplied for the “intensity” of the PTM, the values were summed, and the result divided by the total sum of all intensity values of the respective unmodified peptide sequence, obtaining a deamidation rate between 0 and 1 for each unique peptide sequence and charge state. For each peptide was calculated an average deamidation rate for Asparagine and Glutamine. The deamidation rates were averaged per sample. The latter set of values was sampled with replacement (bootstrapped) 1000 times. The mean, the standard deviation, and the 95% confidence intervals were calculated in order to achieve an estimate of the error of the calculation [30].

The program generates four delimited text files as output: (1) Deamidation.txt (Raw Files, deamidation for N and Q, as mean, standard deviation, 95% confidence lower and upper limit); (2) Number_of_Peptides_per_RawFile.txt; (3) Bootstrapped_values.txt (all the deamidation percentages calculated by e.g., 1000 bootstrap iterations, which are subsequently used to calculate the mean, std, and CI for shown in “Deamidation.txt”); (4) Protein_deamidation.txt (deamidation on the protein level).

#### 2.5.2. Semi-Quantitative Analysis

A semi-quantitative analysis was performed by selecting the Fast LFQ (label-free quantification) option in MaxQuant 1.6.3.4. As recommended by Cox J. [31], “unique plus razor peptides” were included in the quantification. Razor-peptides are non-unique peptides assigned according to Occam’s razor principle. Only unmodified peptides were used for quantification. “Advanced ratio estimation”, “stabilize large LFQ ratios”, “require MS/MS for comparisons” and “advanced site intensities” were selected. The runs were automatically aligned.

### 2.6. Modelling and Bioinformatics Analysis

Computational representation of the structures of VDAC1 N37D, N106D, N207D, N214D, N239D, 5Asn/Asp and 5Asn/Asp-2Gln/Glu were obtained by MODELLER v9.24 software (Webb & Sali, 2016) using the crystallographic structure of mouse VDAC1 WT (PDB: 3EMN) as template. Mutations were introduced by substitution of the selected amino acid residue(s) in the FASTA sequence. The same software was used for the evaluation of the energetic score associated with each structure. Graphical representation was obtained by using VMD—Visual Molecular Dynamic software (available at: https://www.ks.uiuc.edu/Research/vmd/). The root-means-square deviation (RMSD) analysis and the Ramachandran plots (RPs) were obtained by using specific tools in the VMD software. The number of α-helix, β-strand or unstructured domains was then analyzed and expressed as a percentage of the total.

## 3. Results

Our study was mainly focused on the identification of enzymatic and non-enzymatic PTMs in the VDAC1 protein from an established ALS mouse motor neuron-like cell line, especially considering the changes induced by oxidative stress typical of neurons affected by the disease.

Exactly, we searched, by mass spectrometry analysis, PTMs as oxidation and succination of cysteine residues, oxidation of methionine residue, phosphorylation of serine, tyrosine and threonine residues, ubiquitin and ubiquitination of lysine residues, and deamidation of asparagine and glutamine residues (Appendix A).

To exclude the possibility that any unspecific and undesired oxidation could arise from the sample purification procedure, the reduction and alkylation of proteins were performed before VDACs purification. Hydroxyapatite (HTP) eluates of Triton X-100 extract were digested in-solution by trypsin and chymotrypsin and subsequently analysed in triplicate by liquid chromatography-mass spectrometry. In this experiment, every protein in the HTP eluate was digested, thus a very complex peptide mixture was produced. Irrespective of the cell line from which the protein was isolated, nUHPLC/nESI-MS/MS analysis of the in-solution tryptic and chymotryptic digest of VDAC1 allowed to ascertain that the *N*-terminal methionine reported in the SwissProt database sequence (Acc. N. Q60932) was missing in the mature protein (Figure 1) analogously to the rat and human VDAC1 isoform [22,28]. Combining the results obtained in the complementary tryptic and chymotryptic digestions of the three cellular lines, a coverage of 96.8% of VDAC1 sequence was obtained (273 out of 282 amino acid residues; Figure 1). The regions not covered correspond to short traits or single amino acids (Thr^19^-Lys^20^, Leu^29^, Arg^63^, Gly^220^-Lys^224^). MS analysis was also performed on VDAC1 samples purified from another series of NSC34, NSC34-SOD1WT, and NSC34-SOD1G93A cell cultures used in this work, in the same way as the first one. The data obtained were largely superimposable with the former (data not shown).

The sequence of VDAC1 from Mus musculus contains one methionine in position 155, and two cysteines in position 127 and 232. It should be noted that the numeration adopted in the discussion starts from Met^1^, which is absent in the mature protein.

### 3.1. Mass Spectrometry Analysis of VDAC1 from NSC34 Cell Line

The mass spectrometry analysis was first performed on material obtained from NSC34 wild-type cells, to establish a reference pattern and to compare the results with those reported for other cell lines.

Met^155^ was identified in the normal form (Appendix A, fragment 10), but also as Met sulfoxide and Met sulfone (Appendix A, fragments 2 and 3). Appendix A shows the full scan and fragment ion mass spectrum of the molecular ion of the peptide G^140^ALVLGYEGWLAGYQMNFETSK^161^ with Met^155^ oxidized to methionine sulfoxide. The full scan mass spectrum of the molecular ion of the same peptide with Met^155^ oxidized to methionine sulfone is reported in Appendix A. The MS/MS spectrum of Met sulfoxide presents the characteristic neutral loss of 64 Da corresponding to the ejection of methanesulfenic acid from the side chain of MetO [32].

Although from these data a precise determination of the amount of Met, Met sulfoxide, and Met sulfone cannot be obtained, a rough estimation of their relative abundance can be derived from the comparison of the absolute intensities of the multiply charged molecular ions of the respective peptides. From these calculations, a ratio of about 5:1 MetO/Met and 0.1:1 MetO_2_/Met was found (Appendix A).

Regarding the oxidation state of cysteines, mass spectral analysis indicated that Cys^232^ was completely in the carboxyamidomethylated form (Appendix A, fragment 15), while Cys^127^ was exclusively in the oxidized form of sulfonic acid (Appendix A, fragment 1), thus reproducing the same data obtained with the homologous human isoform [31]. Chymotryptic digestion was also performed and the data obtained confirmed the complete trioxidation of Cys^127^ (Appendix A, fragment 1) and the total carboxyamidomethylation of Cys^232^ (Appendix A, fragment 30).

### 3.2. Mass Spectrometry Analysis of VDAC1 from NSC34-SOD1WT Cell Line

In this section VDAC1 modifications found in NSC34 cells expressing a wt form of human SOD1 were analyzed. MS results indicate that the oxidation state of Met and Cys residues are very similar to the one reported for the same isoform from the NSC34 non transfected cells, discussed previously. In particular, in the tryptic digest Met^155^ was detected both in the normal form (Appendix A, fragment 10) and oxidized to sulfoxide and sulfone, (Appendix A, fragments 2 and 3) with a ratio of about 3:1 and 0.1:1 MetO/Met and MetO_2_/Met, respectively (Appendix A). In addition, Cys^127^ was detected only in the form of sulfonic acid (Appendix A, fragment 1), whereas Cys^232^ was fully carboxyamidomethylated (Appendix A, fragment 15).

These results were again confirmed by the complementary chymotryptic digestion (Appendix A, fragment 1, and Appendix A, fragments 36 and 37).

### 3.3. Mass Spectrometry Analysis of VDAC1 from NSC34-SOD1G93A Cell Line

In this section VDAC modifications found in NSC34 cells expressing the G93A mutated sequence of the human SOD1 were analyzed. MS analysis of tryptic and chymotryptic digest of VDAC1 from NSC34-SOD1G93A cell revealed the presence of Cys^232^ exclusively in the carboxyamidomethylated form (Appendix A, fragment 15; Appendix A, fragment 25), and the partial oxidation of Met^155^ to methionine sulfoxide and sulfone (Appendix A, fragments 2 and 3, respectively). In contrast to the results found in other cells, a higher amount of methionine sulfoxide and sulfone was observed (Ox/Red ratio 61:1 and 4.7:1, respectively; Table 1). Furthermore, peptides containing Cys^127^ residue in the carboxyamidomethylated form (Appendix A, fragment 9) and also oxidized to sulfonic acid were detected (Appendix A, fragment 1) with a ratio of 24:1 (Table 1).

### 3.4. Post-Translational Modifications Found in VDAC1 HTP-Prepared from Various NSC34 Cell Lines

#### 3.4.1. Phosphorylation, Succination, Ubiquitin and Ubiquitination, and N-Terminal Acetylation

Mass spectral data were also analyzed to reveal the occurrence of serine, threonine, or tyrosine phosphorylation. In all the replicates of the three cellular lines, only Ser^104^ was observed to be phosphorylated (Appendix A), even if always in a low amount (Appendix A), as indicated in the corresponding MS/MS spectra, which show typical losses of phosphoric acid (P) from fragment ions *y*, in accord with previous reports [33,34] (Appendix A).

The N-end Ala^2^ is only present in acetylated form in all the replicates of the three cell lines (Appendix A, fragment 1; Appendix A, fragment 1; Appendix A, fragment 1 respectively). These results were confirmed by the analysis of the chymotryptic digest (Appendix A, fragments 1 and 2; Appendix A, fragments 1 and 2; Appendix A, fragment 1). Moreover, succinated cysteines were not found in VDAC1 isoforms from any kind of studied NSC34 cell, as well as no evidence of ubiquitin and ubiquitination was detected.

#### 3.4.2. Identification of VDAC1 Deamidation Sites from NSC34-SOD1G93A Cell Line

Deamidations of specific residues of asparagine and glutamine were detected only in VDAC1 purified from NSC34-SOD1G93A cells. This result was confirmed in each replicate analyzed (Appendix A). Exactly, five asparagines (Asn^37^, Asn^106^, Asn^207^, Asn^214^, and Asn^239^, Appendix A, fragments 1, 2, 3, 5, 6, and 8, and Figure 2 and Appendix A) were transformed to aspartate and two glutamines (Gln^166^ and Gln^226^, Appendix A, fragments 4 and 7, and Appendix A) to glutamate.

In particular, Table 2 suggests that an appreciable amount of Asn^207^ is deamidated (ratio deam/norm 0.6), whereas deamidated Gln^166^ and Gln^226^ are visible in trace amounts (ratio deam/norm 0.002). Furthermore, for the other asparagines a ratio Asn-deamidated-to-Asp/Asn ranging from 0.01:1 to 0.04:1 was observed.

The occurrence of deamidation only in the NSC34-SOD1G93A cell line was also confirmed by the calculation of the percentage of deamidation in the VDAC1 for each cell line (Table 3). A remarkable difference in the deamidation between asparagine (mean 0.98) and glutamine residues (mean 0.02) was also found.

### 3.5. Semi-Quantitative Analysis of VDAC1 in NSC34 Cell Lines

The LFQ intensities obtained from MaxQuant search allowed for an estimation of the percentage of each VDAC in every cell line (Table 4). All three cell lines presented VDAC1 as the most abundant isoform (one order of magnitude higher than the other isoforms). However, NSC34 and NSC34-SOD1WT cell lines showed a similar profile for all the isoforms, whereas the NSC34-SOD1G93A cells had a difference in the abundance of VDAC2 and VDAC3. VDAC2 levels were higher in NSC34-SOD1G93A cells compared to NSC34 and NSC34-SOD1WT cell lines. In contrast, VDAC3 levels were lower in NSC34-SOD1G93A cells compared to NSC34 and NSC34-SOD1WT cell lines. The high complexity of the peptide mixtures could affect the reproducibility of the analysis, explaining the high standard deviation in NSC34-SOD1WT cells. Hence, caution about these results is required, although they may represent an interesting hint for further investigations.

### 3.6. Bioinformatic Analysis Reveals an Increased Instability for Mutant Structures

To evaluate the stability of VDAC1 mutants, the mouse crystal structure from PDB [35] was modified with substitution of one or more asparagine or glutamine with aspartate or glutamate, as “virtual” deamidated amino acids. Root-mean-square deviation (RMSD) between the unmodified structure and those carrying such deamidations was analyzed. RMSD measures the distance between the atoms in the backbone and those of a reference structure in a protein structure alignment. All the generated mutant structures were superimposed to VDAC1 wt obtaining in all cases high RMSD values. In particular, for the single mutants, the average RMSD values were 5.16 Å for N37D, 5.28 Å for N106D, 5.17 Å for N214D, 4.79 Å for N239D and 5.20 Å for N207D. Similar values were obtained for multiple mutants (5.33 Å and 5.59 Å for 5 Asn/Asp and 5 Asn/Asp-2 Glu/Gln, respectively).

Overall, these results indicate that mutations produced a substantial modification in VDAC1 structure, likely affecting its molecule stability. Then, Ramachandran plots (RPs) were generated to determine φ and ψ angles for each amino acid residue (Table 5). This information, indeed, is very helpful for structure predictions since it describes the specific orientation of each residue in space. For example, RP for VDAC1 WT indicates that majority of amino acid residues are arranged in a β-strand structure (about 78%) while the rest is distributed between α-helix (8.7%) and outlier conformations (12.3%).

As reported in Table 5, the presence of single mutations specifically affects the distribution of the secondary structures. However, most of the differences were observed for multiple mutants in which, in front of a slight decrease of β-strand structures, the percentage of α-helix increased proportionally to outlier reduction. Structural modifications produced by amino acid substitutions were confirmed by molecular modelling analysis (Figure 3, Figure 4 and Figure 5). In Figure 3 and Figure 4, structural reconstructions of the crystal and mutated sequences are shown. The models visually confirm the results obtained by RMSD calculations and outlined in the secondary structure content determined by the RPs.

The deamidation of the selected asparagine and glutamine causes a crushing of the barrel (see Figure 4c, for example) and a shift of the global secondary structure composition from β-strand to α-helix. What is important is that even a single deamidation has a major structural effect (Figure 3 and Figure 4).

It is interesting that the amino acids found deamidated in the cell expressing the mutated SOD1 are mostly located in loops or turns connecting the β-strands: they are, thus, in a water-exposed context where the effect of ROS can be faster. Only Asn^37^ and Gln^226^ protrude in the intermembrane space, while Asn^106^, Asn^214^, Asn^239^ and Gln^166^ protrude in the cytosol. The only exception is Asn^207^ which is located in the middle of a β-strand, even though it is exposed to the water interior of the pore, thus still accessible to ROS by diffusion. Interestingly, Asn^207^ is also the most frequently deamidated residue (see Table 6) and its modification results in a sensible change of the structure (Figure 3b and Figure 4b), indicating a crucial function for this residue.

## 4. Discussion

The impairment of the bioenergetics metabolism strongly increases free radicals and oxidative stress, which, in turn, are involved in ALS onset and progression. Reactive species damage proteins by inducing various oxidative PTMs some of which are biological markers of misfolding disorders, including Parkinson’s disease, Alzheimer’s disease, and ALS [36,37,38]. In particular, carbonylation of spinal cord proteins has been proposed as a promoting factor of neuronal death in ALS [39,40]. Further reports showed the importance of other PTMs in ALS-related proteins. In particular, SOD1 mutants exhibit structural instability and propensity to exist in an unfolded state as well as presenting specific secondary post-translational changes [41]. Interestingly, by altering the normal set of PTMs [42,43] and/or inserting mutations which result in the same chemical structure as the PTM deamidation (N→D; Q→E), the wild type SOD1 becomes unstable and misfolds, acquiring properties similar to those of fALS-related SOD1 mutants [44]. Therefore, certain PTMs are responsible for important chemical and biophysical characteristics of mutant SOD1 and may be involved in sporadic ALS. Equally relevant to understanding the role of mitochondrial dysfunction in SOD1 linked-ALS are the non-inheritable PTMs of VDAC1, an outer mitochondrial membrane (OMM) scaffold protein able to interact with ALS-linked SOD1 mutants but not with SOD1 wt.

Evidence of oxidative damage to proteins, lipids and DNA has been found both in animal ALS models expressing mutated human SOD1 and in post-mortem tissues of patients with sporadic or familial ALS [7]. In particular, deamidation events were found in erythrocytes of sALS patients [45] and in other tissues from patients with ALS-SOD1G93A [44]. Specific PTM, including N-D e Q-E deamidation, were also found in TDP-43 aggregates from brains of ALS patients [46]. There is no report in the literature where deamidation PTMs were looked for in animals.

In this study, by high-resolution mass spectrometry analysis, possible signs of oxidative damage in VDAC1 from an ALS cell model were investigated. Using known techniques [22], difficulties associated with the analysis of membrane proteins were overcome and potential dangers of undesirable oxidation were avoided. We have achieved excellent VDAC1 sequence coverage from mitochondria isolated from NSC34 cell lines; 273 out of 282 amino acids in the VDAC1 sequence were identified (Figure 1). The non-covered regions corresponded to a few single amino acids and a short stretch (G220–A224) found in the β-strand 15. We accurately determined the oxidative states of the cysteine and methionine residues of NSC34 mouse VDAC1. VDAC1 contains a single inner methionine, Met^155^, and only two cysteines, Cys^127^ and Cys^232^, located in β-strands 10, 8 and 16 respectively. According to VDAC1’s three-dimensional structure [35,47,48], Met^155^ and Cys^127^ are embedded in the hydrophobic milieu of the OMM, whereas Cys^232^ turns toward the water-filled pore. Mass spectrometry analysis revealed an evolutionarily conserved redox modification pattern of VDAC1 cysteine residues: both in human and rat, Cys^127^ was exclusively found as trioxidized to sulfonic acid and Cys^232^ was fully reduced and carboxyamidomethylated [22,28]. Still, a functional role for these residues has never been demonstrated. Data available so far report that cysteines in VDAC1 do not influence the channel activity [49] and are not involved in protein oligomerization [50].

In our study, only in NSC34-SOD1G93A cells, we found that Met^155^ was oxidized to methionine sulfone and mostly over-oxidized to methionine sulfoxide also in much higher amounts than the one found for the same residue in the two non-ALS cell lines. Regarding the other residues, Cys^232^ was confirmed as completely reduced or involved in forming disulfide bridges in ALS and non-ALS cell lines [28]. Conversely, Cys^127^ from SOD1-ALS cells was over-oxidized to sulfonic acid, in a lower amount than in the two non-ALS cell lines.

Oxidative modifications in both Met^155^ and Cys^127^ could be explained by their localization in neighboring β-sheet where they face the lipid bilayer. Thus, we hypothesize that OMM lipids peroxidized by ALS pathogenesis [51] can modify susceptible residues oriented towards the membrane. A low amount of the Cys^127^ was also in carboxyamidomethylated form, therefore in a reduced form or involved in an intermolecular disulfide bridge (due to cysteines position in VDAC1). The finding of a putatively reduced aliquot of Cys^127^ could be a consequence of the destabilization of the VDAC1 structure following all the modifications we identified. Indeed, we have also revealed only in NSC34-SOD1G93A cells, the deamidation of specific asparagine and glutamine residues. This is the first time that these changes were reported in VDAC1. Deamidation is a non-inheritable PTM that introduces negative charges by removing amino groups from asparagine and, with a much lower frequency and rate, glutamine [52,53]. This modification is strongly associated with the regulation of protein folding and turnover. While in vivo enzymatic deamidation of Gln is known and is involved in several physiological and pathological processes, Asn deamidation is a non-enzymatic reaction and its process rate depends on several factors, including the protein sequence and structure, as well as temperature, ionic strength, and pH. In particular, the propensity to undergo deamidation is considerably increased if an Asn residue is in a protein sequence followed by small, flexible, or hydrophilic amino acids [54]. Several examples of human deamidated proteins are known, most of which are implicated in neurodegeneration [44,55,56,57].

As it causes critical alterations in peptide and protein structure, deamidation is suspected to contribute to the aging of proteins and protein folding disorders such as those observed in Alzheimer’s disease [56,57,58]. It has also been demonstrated that specific deamidations of SOD1 wt determine aberrant conformations in the protein similar to those of the mutated forms associated with fALS [43,59].

In this work, we found the surprising result of identifying deamidation of five asparagine (Asn^37^, Asn^106^, Asn^207^, Asn^214^, and Asn^239^) and two glutamine (Gln^166^ and Gln^226^) residues, exclusively in VDAC1 purified from NSC34-SOD1G93A cell line (Figure 2, Table 2, Table 5 and Table 6). In particular, we found that the deamidation level of the Asn^207^ residue was much higher than other asparagine residues converted to aspartate, and especially that the two glutamines were modified to glutamate in a very small quantity (Table 2 and Table 6). This may depend on the particular position of Asn^207^ in the VDAC1 sequence. In fact, this residue is located in the centre of the β-strand 14 and is preceded and followed by non-bulky amino acid residues, such as the nearby residues in the flanking β-strands (Figure 3 and Figure 4). As a result of this, the amido group of Asn^207^ can be particularly sensitive to ROS at the inner mitochondrial membrane (IMM) which are strongly increased in ALS, as well as being affected by the aggressive action of peroxidized lipids in the OMM. The lower level of deamidation of the Asn^37^ residue could depend on its position in a loop, a domain that, although facing the oxidizing environment of the IMM, constitutes an unfolded region and therefore it is free to move and/or establish interactions with other molecules as well as to protect the Asn^37^ from the action of ROS (Figure 3 and Figure 4). The other three asparagine residues (Asn^106^, Asn^214^ and Asn^239^) are also found in the loops (Figure 3 and Figure 4), but unlike Asn^37^ these loops are exposed to the cytosol, a much less oxidizing environment than IMM.

We are sure that the different levels of oxidation and deamidation of VDAC1 identified in this study were not due to different protein expression in ALS motor neurons. In fact, an equal expression of VDAC1 was found in the three cell lines analysed, while VDAC2 levels were increased only in the ALS model NSC34 (Table 4). This result could be explained by considering the anti-apoptotic role of VDAC2 isoform, which could answer to degeneration signs characteristic of ALS. Deamidation introduces a negative charge in the protein sequence and, as a result, is often responsible for a serious change in its structure. RMSD analysis performed on the predicted structures for deamidated mouse VDAC1 showed that deamidation undoubtedly modifies the VDAC1 protein structure.

Structural predictions obtained by molecular modelling support data from RMSD analysis. Interestingly, conformational changes in deamidated VDAC mutants were found. In particular, it should be noted that the Asn^207^/Asp^207^ replacement alone causes structural modifications to the VDAC1 channel that becomes more and more marked as the number of deamidations increases (Figure 3 and Figure 4). The distribution of the charges in the structures of the deamidated VDAC1 is also remarkably interesting: the replacement of the five asparagines and the two glutamines with the same number of aspartates and glutamates, respectively, increases the number of negative charges on the pore surfaces facing the cytosol and the intermembrane space (Figure 5). This different charge distribution must have an impact on the physiological interactivity of VDAC1 and consequently on the functionality of the protein itself and of the whole mitochondrion.

Moreover, in the predicted structures we have also observed that Met^155^ and Cys^127^ in deamidated VDAC1 maintain their original arrangement, although the sulphur of Met^155^ is further exposed to the lipid environment. In addition, we found that the atomic distance between Cys^127^ and the nearby residues was for Leu^125^ decreased from 10.2 A in VDAC1 WT to 8.6 A in deamidated VDAC1, shielding on one side the Cys^127^ in the sulfonic acid form (data not showed). These data may explain the small amount of Cys^127^ in reduced form found in ALS model cells.

RMSD data were confirmed by Ramachandran plot (RP) analysis for each single deamidated VDAC form identified in this study. Structural modifications were detected in the β-barrel regions of VDAC1 mutants together with an increased number of residues in the a-helix region, indicating a tendency for this domain to organise itself into a more stable structure. This effect, which causes destabilisation of the whole VDAC1 structure and pushes towards alpha helix formation, already occured with the N207D mutant or any other evaluated single mutant. Interestingly, RPs show that this trend increase with the rise of deamidation sites. It is reasonable to believe that the different levels of deamidation of the residues identified in VDAC1 may depend on the degree of disease that NSC34-SOD1G93A cells reflect. It is conceivable that the level of deamidation of VDAC1 will gradually increase with the worsening of the disease (Figure 6). D’Angelo S. et al. [45] reported the abnormal deamidation of protein asparagine residues in erythrocytes from sALS patients. In this perspective, the search for deamidated VDAC in blood or cerebrospinal fluid could represent an innovative diagnostic and/or prognostic biomarker of amyotrophic lateral sclerosis.

## 5. Conclusions

In this work, we have established for the first time that in a motor neuron model of ALS, VDAC1 undergoes specific post-translational modifications, such as oxidations and deamidations. In addition, we have provided evidence of changes in the structure of deamidated VDAC1, a condition sufficient to alter the physiological pool of interactors. These data are important to understand how the VDAC1 protein interacts uniquely with the mutant SOD1 mutant but also allows us to better understand ALS pathogenesis. Furthermore, a new role as a potential marker for degenerated and unrepairable mitochondria in ALS emerges for VDAC1.

## Figures and Tables

**Figure 1 antioxidants-09-01218-f001:**
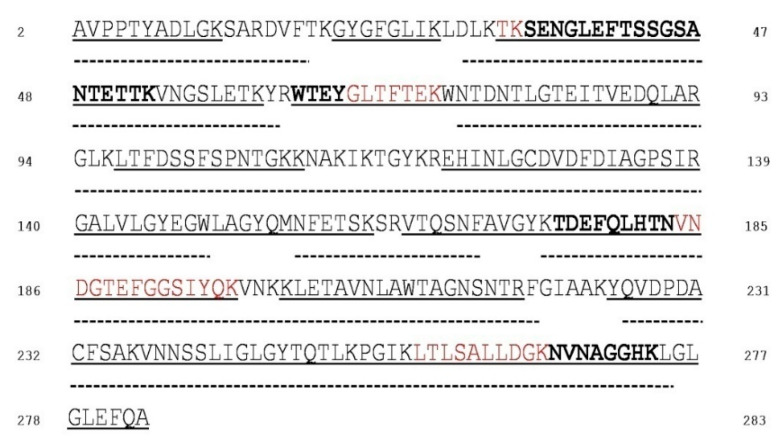
Sequence coverage map of VDAC1 from NSC34, NSC34-SOD1WT and NSC34-SOD1G93A cell lines obtained by tryptic and chymotryptic digestion. The solid line indicates sequence that was obtained from tryptic peptides; dotted lines: sequence obtained from chymotryptic peptides. Unique tryptic (indicated in bold) peptides originated by missed-cleavages were used to distinguish and cover sequences shared by isoforms. Sequences shared by more isoforms are reported in red. The numbering of the sequence considered the starting methionine residue, which is deleted during protein maturation.

**Figure 2 antioxidants-09-01218-f002:**
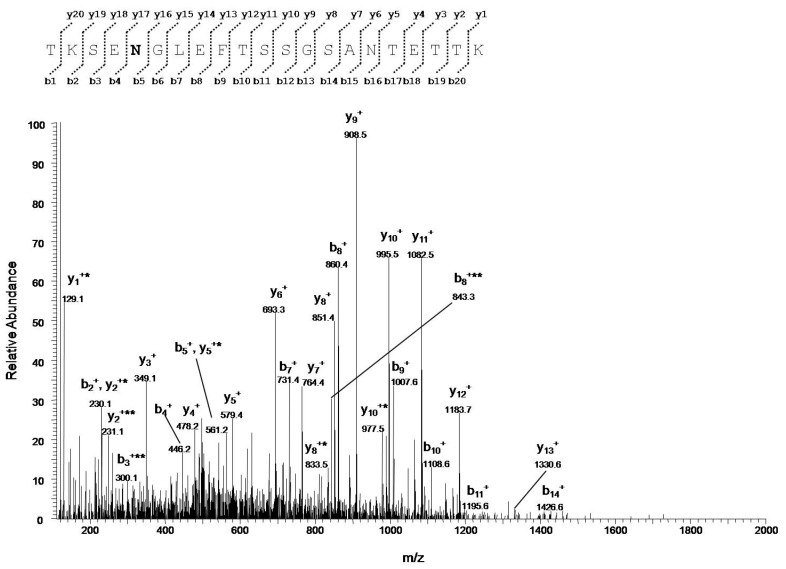
MS/MS spectrum of the triply charged molecular ion at *m/z* 730.6730 (calculated 730.6734) of the VDAC1 tryptic peptide from NSC34-SOD1G93Acellcontaining Asn residue 37 in the deamidated form. Fragment ions originated from the neutral loss of H_2_O are indicated by an asterisk. Fragment ions originated from the neutral loss of NH_3_ are indicated by two asterisk.

**Figure 3 antioxidants-09-01218-f003:**
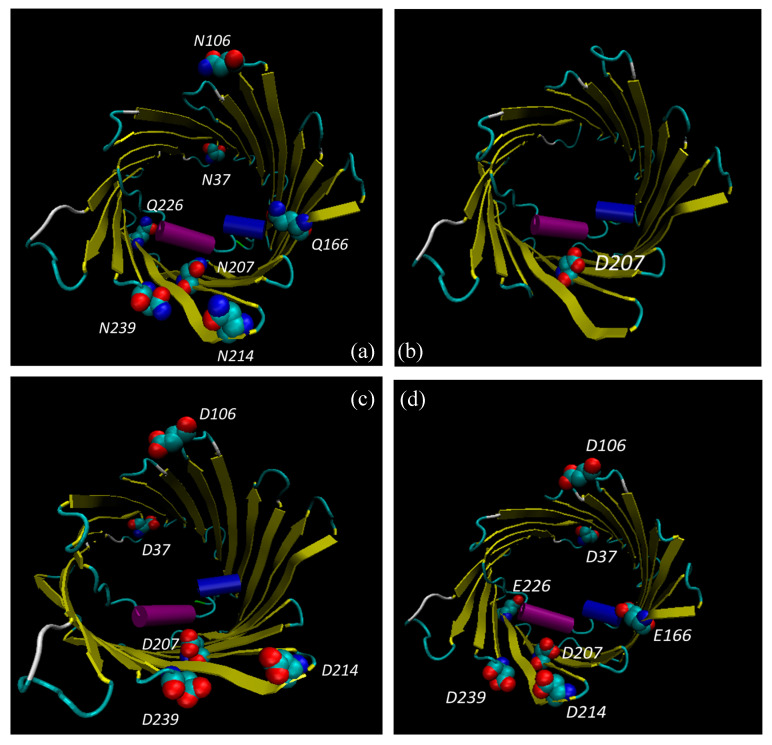
Top view of VDAC1 structures. Graphic representation of VDAC1 structures obtained by modelling WT (PDB: 3EMN) (**a**), mutant N207D (**b**), deamidated mutant 5 Asn/Asp (**c**) and deamidated mutant 5 Asn/Asp 2 Gln/Glu (**d**). N-terminal α-helix is shown in purple, β-strands in yellow, unstructured loops/regions in white and light blue. The amino acid residues Asn, Gln and Asp are evidenced. Dark blue was used for nitrogen, red for oxygen and light blue for carbon atoms.

**Figure 4 antioxidants-09-01218-f004:**
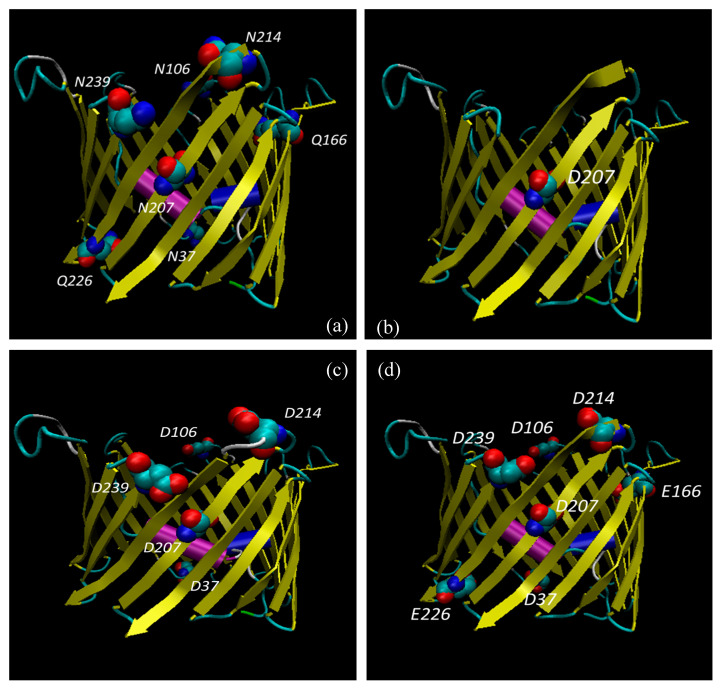
Lateral view of VDAC1 structures. Graphic representation of VDAC1 structures obtained by modelling of WT (PDB: 3EMN) (**a**), mutant N207D (**b**), deamidated mutant 5 Asn/Asp (**c**) and deamidated mutant 5 Asn/Asp 2 Gln/Glu (**d**). N-terminal α-helix is shown in purple, β-strands in yellow, unstructured loops/regions in white and light blue. The amino acid residues Asn, Gln and Asp were evidenced. Dark blue was used for nitrogen, red for oxygen and light blue for carbon atoms.

**Figure 5 antioxidants-09-01218-f005:**
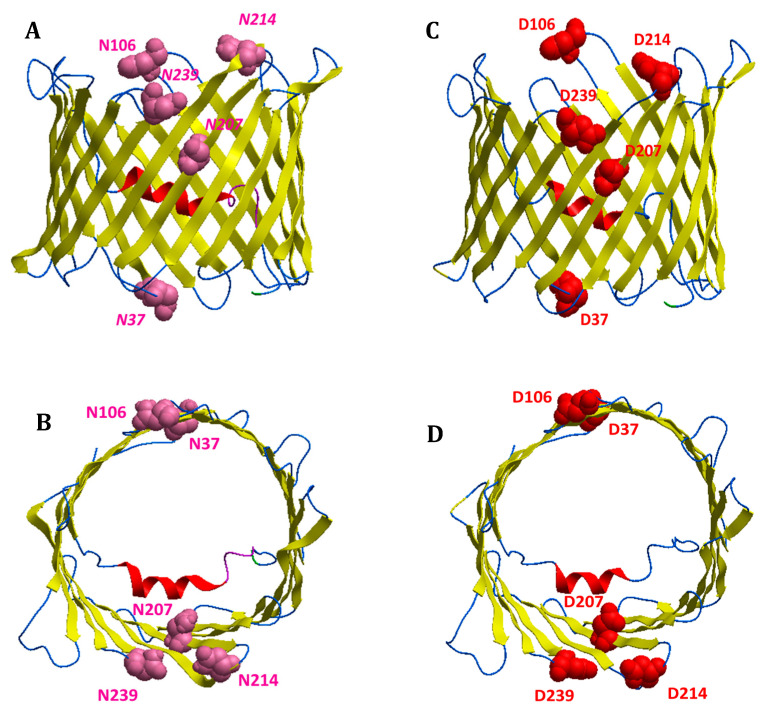
Polarity changes upon VDAC1 deamidation. Graphic representation of VDAC1 structures obtained by modelling in which polarity of specific amino acid residues has been highlighted. Lateral and top view of VDAC1 WT (**A**,**C**) and 5 Asn/Asp deamidated mutant (**B**,**D**). Pink color is used for amino acid residues having a neutral lateral chain, while red color is used for negatively charged residues.

**Figure 6 antioxidants-09-01218-f006:**
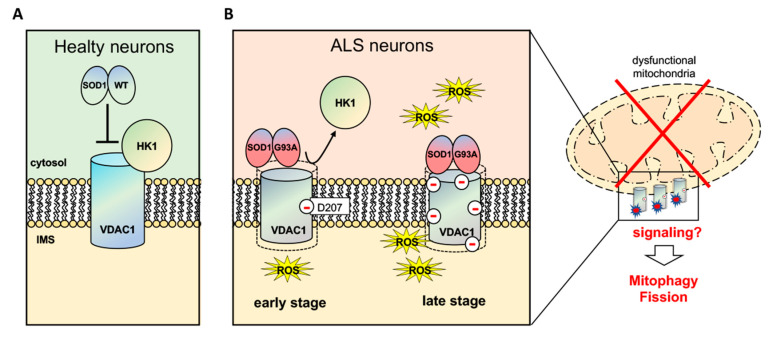
Deamidated VDAC1 in ALS motor neurons can promote mitochondrial dysfunction and ultimately lead to mitophagy. In physiological conditions (**A**), VDAC1 is the receptor of HK1 and of many other enzymes and metabolites (not shown), but not of SOD1WT. Conversely, in ALS (**B**), SOD1G93A is able to bind deamidated VDAC1 impairing thus the binding of HK1 and likely of others physiological interactors. We propose that deamidation of VDAC1 residues parallels oxidative stress levels, leading to the enhancement of SOD1 mutant binding to VDAC1. As a consequence, VDAC1 channel conductance and metabolic exchanges through VDAC1 are gradually affected, promoting a growing mitochondrial dysfunction. The oxidative damage that accumulates with disease progression produces therefore dysfunctional mitochondria tagged with deamidated VDAC1 that could be selected and forwarded to mitophagy or fission processes. In figure, red minus signs indicate deamidated VDAC1 amino acids.

**Table 1 antioxidants-09-01218-t001:** Ratio of the absolute intensities of the molecular ions of the sulfur-containing tryptic peptides found in the analysis of VDAC1 from NSC34-SOD1G93A cells HTP-preparation reduced with DTT, carboxyamidomethylated and digested in-solution.

Technical Replicate	Peptide	Position in the Sequence	Measured Monoisotopic *m/z*	Absolute Intensity	Ratio Ox/Red
I	EHINLG**C**DVDFDIAGPSIR	121–139	1059.9921 (+2)	8.2 × 10^5^	29.3
I	EHINLG***C***DVDFDIAGPSIR		710.0155 (+3)	2.8 × 10^4^	
II	EHINLG**C**DVDFDIAGPSIR		1059.9915 (+2)	6.9 × 10^5^	19.2
II	EHINLG***C***DVDFDIAGPSIR		710.0140 (+3)	3.6 × 10^4^	
III	EHINLG**C**DVDFDIAGPSIR		1059.9928 (+2)	7.8 × 10^5^	23.6
III	EHINLG***C***DVDFDIAGPSIR		710.0140 (+3)	3.3 × 10^4^	
I	GALVLGYEGWLAGYQ**M**NFETSK	140–161	1225.5892 (+2)	4.5 × 10^6^	60.0
I	GALVLGYEGWLAGYQMNFETSK		1217.5931 (+2)	7.5 × 10^4^	
II	GALVLGYEGWLAGYQ**M**NFETSK		1225.5897 (+2)	4.9 × 10^6^	71.0
II	GALVLGYEGWLAGYQMNFETSK		1217.5913 (+2)	6.9 × 10^4^	
III	GALVLGYEGWLAGYQ**M**NFETSK		817.3946 (+3)	2.8 × 10^6^	51.9
III	GALVLGYEGWLAGYQMNFETSK		812.0656 (+3)	5.4 × 10^4^	
I	GALVLGYEGWLAGYQ**M**NFETSK	140–161	822.7261 (+3)	2.2 × 10^5^	2.9
I	GALVLGYEGWLAGYQMNFETSK		1217.5931 (+2)	7.5 × 10^4^	
II	GALVLGYEGWLAGYQ**M**NFETSK		822.7269 (+3)	4.5 × 10^5^	6.5
II	GALVLGYEGWLAGYQMNFETSK		1217.5913 (+2)	6.9 × 10^4^	
III	GALVLGYEGWLAGYQ**M**NFETSK		822.7277 (+3)	2.6 × 10^5^	4.8
III	GALVLGYEGWLAGYQMNFETSK		812.0656 (+3)	5.4 × 10^4^	

***C***: cysteine carboxyamidomethylated; **C**: cysteine oxidized to sulfonic acid; **M**: methionine sulfoxide; **M**: methionine sulfone.

**Table 2 antioxidants-09-01218-t002:** Ratio of the absolute intensities of the molecular ions of tryptic peptides containing deamidated against not deamidated amino acids, found in the analysis of VDAC1 from NSC34-SOD1G93A cell HTP-preparation reduced with DTT, carboxyamidomethylated and digested in-solution.

Peptide	Position in the Sequence	Measured Monoisotopic *m/z*	Absolute Intensity	Ratio Deam/Norm
TKSE**N**GLEFTSSGSANTETTK	33–53	730.6730 (+3)	6.3 × 10^6^	0.02
TKSENGLEFTSSGSANTETTK		730.3450(+3)	3.3 × 10^8^	
SE**N**GLEFTSSGSANTETTK	35–53	654.2921 (+3)	6.3 × 10^5^	0.01
SENGLEFTSSGSANTETTK		980.4425(+2)	1.4 × 10^8^	
LTFDSSFSP**N**TGKK	97–110	510.5876 (+3)	1.3 × 10^6^	0.01
LTFDSSFSPNTGKK		764.8857 (+2)	9.9 × 10^7^	
VT**Q**SNFAVGYK	164–174	607.8062 (+2)	1.4 × 10^6^	0.002
VTQSNFAVGYK		607.3142 (+2)	6.2 × 10^8^	
KLETAV**N**LAWTAGNSNTR	201–218	649.6692 (+3)	9.6 × 10^6^	0.6
KLETAVNLAWTAGNSNTR		649.3412 (+3)	1.6 × 10^7^	
LETAVNLAWTAG**N**SNTR	202–218	909.9527(+2)	6.8 × 10^5^	0.04
LETAVNLAWTAGNSNTR		909.4607 (+2)	1.9 × 10^7^	
Y**Q**VDPDA*C*FSAK	225–236	701.3032(+2)	1.6 × 10^6^	0.002
YQVDPDA*C*FSAK		700.8112(+2)	7.1 × 10^8^	
VN**N**SSLIGLGYTQTLKPGIK	237–256	702.0598 (+3)	2.4 × 10^6^	0.03
VNNSSLIGLGYTQTLKPGIK		701.7318 (+3)	7.4 × 10^7^	

***C***: cysteine carboxyamidomethylated; **N**: asparagine deamidated; **Q**: glutamine deamidated.

**Table 3 antioxidants-09-01218-t003:** Asparagine (N) and glutamine (Q) deamidation in the VDAC1 from NSC34 cells HTP-preparations. Mean, standard deviation, (95%) lower and upper confidence level and the number of scansions are reported.

Cell Lines	Residue	Mean	Std	CI_Low	CI_Up	N. of Scansions
**NSC34**	N	0	0	0	0	51
	Q	0	0	0	0	30
**NSC34-SOD1WT**	N	0	0	0	0	61
	Q	0	0	0	0	41
**NSC34-SOD1G93A**	N	0.98	0.81	0.04	2.76	87
	Q	0.02	0.02	0.00	0.06	47

(95%) lower and upper confidence level were obtained from 1000 bootstrap.

**Table 4 antioxidants-09-01218-t004:** Relative percentage of each VDAC isoform in the three cell lines.

Protein	NSC34			NSC34-SOD1WT			NSC34-SOD1G93A		
	LFQ Intensity	SD%	%	LFQ Intensity	SD %	%	LFQ Intensity	SD %	%
**VDAC1**	1.10 × 10^9^	1	66.71	1.19 × 10^9^	27	65.67	1.10 × 10^9^	7	64.08
**VDAC2**	2.40 × 10^8^	4	14.55	2.46 × 10^8^	20	13.59	3.51 × 10^8^	11	20.43
**VDAC3**	3.09 × 10^8^	15	18.74	3.75 × 10^8^	16	20.74	2.66 × 10^8^	13	15.50

f.e. VDAC1(%) = mean(LFQ Intensity)_(VDAC1)_/[mean(LFQ Intensity)_(VDAC1)_ + mean(LFQ Intensity)_(VDAC2)_ + mean(LFQ Intensity)_(VDAC3_].

**Table 5 antioxidants-09-01218-t005:** Distribution of secondary structures as predicted by Ramachandran Plot analysis. The table reports the percentage of allowed, favored and total secondary structure consisting of outliers, β-strand or α-helix, as obtained by RP analysis of VDAC1 WT and mutant structures.

	Outliers	β-Strand			α-Helix		
	Total	Allowed	Favored	Total	Allowed	Favored	Total
VDAC1 WT	12.4	11.3	67.6	78.9	6.3	2.4	8.7
VDAC1 N37D	10.6	13.5	63.2	76.7	7.8	4.9	12.7
VDAC1 N106D	8.8	12.8	65.4	78.2	7.0	6.0	13.0
VDAC1 N207D	9.1	13.6	64.6	78.2	7.4	5.3	12.7
VDAC1N214D	7.7	12.3	66.1	78.4	6.7	7.1	13.8
VDAC1N239D	8.1	12.3	66.4	78.7	7.1	6.1	13.2
VDAC1 5 Asn/Asp	7.3	16.6	60.7	77.3	8.4	7.0	15.4
VDAC1 5 Asn/Asp + 2 Gln/Glu	7.7	14.4	60.7	75.1	8.4	8.4	16.8

**Table 6 antioxidants-09-01218-t006:** Modified residues found in VDAC1 from NSC34, NSC34-SOD1WT and NSC34-SOD1G93A cell lines. The ratio between modified and unmodified residues is shown in brackets; where the ratio is not indicated the residue is only present in a single form. The dashes indicate residues found exclusively in non-modified form. The asterisk indicates a fully carboxyamidomethylated form.

	Modified Residues in VDAC1		
	NSC34	NSC34-SOD1WT	NSC34-SOD1G93A
**ALA ^2^**	Acetylated	Acetylated	Acetylated
**ASN ^37^**	-	-	Deamidated (0.01:1)
**SER ^104^**	Phosphorylated (0.01:1)	Phosphorylated (0.01:1)	Phosphorylated (0.01:1)
**ASN ^106^**	-	-	Deamidated (0.01:1)
**CYS ^127^**	Sulfonic acid	Sulfonic acid	Sulfonic acid (24:1)
**MET ^155^**	Met sulfoxide (5:1) Met sulfone (0.1:1)	Met sulfoxide (3:1) Met sulfone (0.1:1)	Met sulfoxide (61:1) Met sulfone (4.7:1)
**GLN ^166^**	-	-	Deamidated (0.002:1)
**ASN ^207^**	-	-	Deamidated (0.6)
**ASN ^214^**	-	-	Deamidated (0.04:1)
**GLN ^226^**	-	-	Deamidated (0.002:1)
**CYS ^232^**	*	*	*
**ASN ^239^**	-	-	Deamidated (0.03)

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
