# Peer review of "Post-Translational Modification Analysis of VDAC1 in ALS-SOD1 Model Cells Reveals Specific Asparagine and Glutamine Deamidation"

_antioxidants, 2020, doi:10.3390/antiox9121218_

Round 1

Reviewer 1 Report

This study used a clever cell line model of ALS to perform mass spec and observe PTMs of VDAC1 in the SOD1 mutant line compared to the SOD1 wt and non-transfected control line.  Since mutant SOD1, but not wt SOD1, directly interacts with VDAC1, the observation of unique PTMs presents a potential pathway for OMM dysregulation, mitochondrial dysfunction and disease pathogenesis.  The work is important, and not only suggests a mechanism of ALS, but may extrapolate to other neurodegenerative diseases caused by mutant aggregation-prone proteins.

I believe the scientific methodology is sound, the results are of interest, and the conclusions resonate directly from the results.  I am in favor of publishing this work.

I would recommend, however, the addition of a cartoon schematic depicting the PTMs and the mechanism of VDAC1-SOD1 interaction for the purpose of clarity.  

Additionally, the manuscript needs extensive revision both for typos and accidental deletions, as well as for English language grammar.  

A few other comments:

-No need to duplicate the Intro at the beginning of the Discussion.  Can remove/modify the first couple paragraphs of the Discussion.

-Some spaces between words are missing.  Likely due to a PDF formatting error, but worth a thorough check.

-Check abbreviations.  If an abbreviation is to be used, include it in parentheses after the first use of the full name, then only use the abbreviation.  Some abbreviations (mt, for instance) could be removed.  Avoid introducing an abbreviation after the full name has been used multiple times.

Author Response

Reviewer #1 (Remarks to the Author):

This study used a clever cell line model of ALS to perform mass spec and observe PTMs of VDAC1 in the SOD1 mutant line compared to the SOD1 wt and non-transfected control line.  Since mutant SOD1, but not wt SOD1, directly interacts with VDAC1, the observation of unique PTMs presents a potential pathway for OMM dysregulation, mitochondrial dysfunction and disease pathogenesis.  The work is important, and not only suggests a mechanism of ALS, but may extrapolate to other neurodegenerative diseases caused by mutant aggregation-prone proteins.

I believe the scientific methodology is sound, the results are of interest, and the conclusions resonate directly from the results.  I am in favor of publishing this work.

I would recommend, however, the addition of a cartoon schematic depicting the PTMs and the mechanism of VDAC1-SOD1 interaction for the purpose of clarity.

We thank very much the reviewer for his/her appreciation. We added Figure 6, a cartoon summarizing the rationale of the work.

- Additionally, the manuscript needs extensive revision both for typos and accidental deletions, as well as for English language grammar. 

The manuscript was revised by an English mother tongue reader.

- A few other comments:

- No need to duplicate the Intro at the beginning of the Discussion. Can remove/modify the first couple paragraphs of the Discussion.

We removed the first couple paragraphs of the Discussion, as suggested by the reviewer.

- Some spaces between words are missing.  Likely due to a PDF formatting error, but worth a thorough check.

We checked PDF formatting to avoid mistakes.

- Check abbreviations.  If an abbreviation is to be used, include it in parentheses after the first use of the full name, then only use the abbreviation.  Some abbreviations (mt, for instance) could be removed.  Avoid introducing an abbreviation after the full name has been used multiple times.

The manuscript was revised for abbreviations, as suggested.

Reviewer 2 Report

Pittalà et al., present an interesting work showing selective deamidations of asparagine and glutamine of VDAC1 in ALS-related NSC34-SOD1G93A cells but not in NSC34-SOD1WT or NSC34 cells. They also find differences in the over-oxidation of methionine and cysteines between VDAC1 purified from ALS model or non-ALSNSC34 cells. The authors conclude that changes in the structure of the VDAC1 channel may affect bioenergetics metabolism of ALS motor neurons (MN).

It is well known that the voltage-dependent anion channel (VDAC) and mitochondrial adenine nucleotide translocator (ANT) are both dimers located in the outer and inner membranes, respectively. High membrane potential in normally functioning mitochondria keeps the mitochondrial permeability pore (PTP) closed. Under pathophysiological conditions such as oxidative/nitrosative stress, the PTP opens, allowing the entry of H2O and solutes. Thus causing mitochondrial swelling and release of apoptosis-initiating factor (AIF) and cytochrome c from the intermembrane space, which ultimately results in apoptosis. ROS can cause e.g. oxidation of critical redox sensitive –SH groups of the ANT (e.g., Cys56), decrease the mitochondrial membrane potential and trigger structural changes leading to the formation of the PTP complex.

Cyclophiline D and hexokinase are required for the formation of the VDAC and ANT pore complex. Cyclophilin D can be deacetylated by SIRT3, which also induces dissociation of hexokinase, and thus may result in inhibition of the PTP formation Moreover, SIRT3 may be inhibited via thiol-specific modification.

It is also well known the importance of ROS and MN mitochondria in the pathophysiology of ALS.

The main problems affecting the present contribution are:

  1. The experimental model. NSC-34 is an hybrid cell line produced by the fusion of MN from the spinal cords of mouse embryos with mouse neuroblastoma cells N18TG2. The selective motor neuronal cell death in ALS is highly dependent on intracellular Ca2+ and is insensitive to inhibitors of voltage-operated Ca2+ and Na+ channels. However, NSC-34 MN-like cells are unsuitable as experimental model for e.g. glutamate-mediated excitotoxicity (doi.org/10.3389/fncel.2016.00118). Therefore, it seems quite clear that the use of primary cultures of MN is more suitable than NSC-34 cell line to explore the pathogenesis. Isolation of MN from mice is a technique which can be used routinely and should have been selected and implemented before starting these experiments.
  2. This work should have analyzed and discuss how changes in the structure of the VDAC1 affect ANT and the formation of the PTP. This is fundamental because involves the mechanism of apoptosis activation as end event in the process of MN degeneration and death.

Author Response

Reviewer #2 (Remarks to the Author):

  1. The experimental model. NSC-34 is an hybrid cell line produced by the fusion of MN from the spinal cords of mouse embryos with mouse neuroblastoma cells N18TG2. The selective motor neuronal cell death in ALS is highly dependent on intracellular Ca2+ and is insensitive to inhibitors of voltage-operated Ca2+ and Na+ channels. However, NSC-34 MN-like cells are unsuitable as experimental model for e.g. glutamate-mediated excitotoxicity (doi.org/10.3389/fncel.2016.00118). Therefore, it seems quite clear that the use of primary cultures of MN is more suitable than NSC-34 cell line to explore the pathogenesis. Isolation of MN from mice is a technique which can be used routinely and should have been selected and implemented before starting these experiments.

We thank the referee for the comment but we do not agree with him/her for the following reasons:

  1. the cellular model of ALS used in our work (as in a previous work published by us in Sci. Reports in 2016) represents a line of motor neuron-like NSC-34 cells that permanently express human mutant G93A SOD1. This cell line, although mixed because it consists of motor neurons and partly neuroblastoma cells, is a recognized cellular model of ALS genetics: several dozen works on ALS can be found on Pubmed that report the use of NSC34 SOD1 G93A or NSC34 expressing other genes related to the disease [e.g. Fiona M. Menzies, et al. Brain (2002) 125: 1522–1533; Hemendinger, R.A., et al. Neurotox res(2008) 13, 49–61; Magrì, A., et al.   Sci Rep(2016) 6, 34802].
  2. the NSC34 used in the work cited by the referee are differentiated cells not expressing the SOD1 G93A mutant (or other protein associated to ALS) and have been used to study glutamate-mediated excitotoxicity; on the contrary, NSC34 used in our work are not differentiated cells and have not been used to study the mechanism of glutamate-mediated excitotoxicity. Therefore, cell model and experimental condition are different.
  3. our recent investigation on VDAC1-SOD1G93A interaction reported in [Magrì, A., et al. Sci Rep(2016) 6, 34802] resembled results obtained in motor neurons isolated from transgenic mice expressing human SOD1G93A. These findings clearly indicate that not differentiated NSC34 are suitable for VDAC analysis. Moreover, since our work merely addresses VDAC PTMs within a disease-like motor neuron model, it must be taken into account what reported in [Chen P. C. et al. Cell Physiol Biochem (2018) 48:2374–2388]. Accordingly, the optimized differentiation of NSC34 cells by using RA affects the activity and the expression of ion channels such as IKC.
  4. undoubtedly primary cultures of motor neurons represent the first choice models of ALS but it is not so easy to have primary ALS model neurons available. In fact, to represent a valid model of the disease, the transgenic ALS mouse should be available; alternatively, primary motor neurons cultures taken from a wt mouse and then transfected with a construct expressing SOD1 G93A should be used. In this last case, however, the transfection efficiency of the primary neurons is usually very low, thus it would associate with a low specific expression and, instead, a heterogeneous expression of SOD1 G93A in the cell culture, making this model difficult to standardize and, thus, absolutely unreliable for our purposes.
  5. it is well known that primary neurons are not a suitable cell model to perform MS analysis because they do not guarantee the amount of sample needed to perform the analysis and provide statistically valid data. It would be even more difficult to obtain suitable amounts of a protein purified from mitochondria!

2. This work should have analyzed and discuss how changes in the structure of the VDAC1 affect ANT and the formation of the PTP. This is fundamental because involves the mechanism of apoptosis activation as end event in the process of MN degeneration and death.

We appreciate the comment, albeit the involvement of VDAC-ANT interaction in permeability transition pore (PTP) is still controversial. Unfortunately only outdated literature proposed an essential role of VDAC in PTP, ) in association with ANT.  (Colombini, M, J. Membr. Biol. (1989) 111: 103-111; Marzo et al., J. Exp. Med. (1998) 187: 1261-1271). Genetic studies showed indeed mPTP activity in the absence of VDAC, suggesting that it is not integral component of the PTP structure, but rather may play regulatory roles in the pore formation [Kokoszka et al., Nature  (2004) 427: 461–465; Krauskopf et al., Biochim. Biophys. Acta (2006) 1757: 590-595; Baines et al., Nat. Cell Biol. (2007) 9: 550-555]. F0F1-ATP synthase dimers are now considered the major constituent of mPTP (Giorgio et al., Proc. Natl. Acad. Sci. U.S.A. (2013) 110: 5887–5892; Carraro et al., J. Biol. Chem. (2014) 289: 15980-15985; Nat. Commun. 2019 Sep 25;10(1):4341; FEBS Lett. 2019 Jul;593(13):1542-1553). In particular, the catalytic site of the F0F1-ATP synthase β subunit constitutes the Ca2+ trigger site that in turn induces a conformational change and transition of the ATP synthase to a channel (Giorgio et al., EMBO Rep. (2017) 18: 859-1037).

Furthermore, our work does not investigate the contribution of the individual pathophysiological mechanisms of ALS in determining the post-translational modifications of the VDAC protein but, considering the specific modifications found and the data obtained from the simulations of the VDAC structure, we hypothesize their possible significance in the pathological context of the disease. However, following the referee's suggestion, we have now briefly discussed in Introduction how modulation of the VDAC channel affects the activation of apoptosis.

Reviewer 3 Report

The paper entitled “Post-translational modification analysis of VDAC1 in ALS-SOD1 model cells reveals specific asparagine and glutamine deamination” is well written, and has merit of publication. However, authors must take in consideration some issues:

- Number of replicates/number used for culture? It should be referred.

-statistical analysis should be strongly supported.

- A resume integrative picture could be done to provide an illustration of data and transpose it.

- Raw data should be submitted via PRIDE or similar.

Author Response

Reviewer #3 (Remarks to the Author):

The paper entitled “Post-translational modification analysis of VDAC1 in ALS-SOD1 model cells reveals specific asparagine and glutamine deamination” is well written, and has merit of publication.

We are happy that the reviewer 3 has appreciated our work and we thank her/him.

However, authors must take in consideration some issues:

- Number of replicates/number used for culture? It should be referred.

We have included the missing data along the paragraph “Methods”.

- statistical analysis should be strongly supported.

 We would like to apologize for not being sufficiently clear. The percentage of deamidation for each cell line was calculated with the aid of a freely available command-line script for Python 2.x, which uses the MaxQuant “evidence.txt” file. This method is currently used for comparative purpose, as reported in reference [32]. To make the text clearer to the reader, we have provided a more detailed description of the procedure.

- A resume integrative picture could be done to provide an illustration of data and transpose it.

We thank the referee for the suggestion: a summary figure (Figure 6) has been added to the manuscript.

 - Raw data should be submitted via PRIDE or similar.

The raw data have been submitted via PRIDE.

Reviewer 4 Report

In this paper, Authors aimed to assess the ability of oxidative stress observed in ALS to induce post-translational modifications (PTMs) in VDAC1, the main protein of the outer mitochondrial membrane. The experimental procedure was performed by using mitochondria of a NSC34cell line expressing human SOD1G93A, a cellular model of ALS and, based on the obtained results, Authors clamed the presence of significant changes to the structure of the VDAC1 channel and to the bioenergetics metabolism of ALS motor neurons.

I found this paper interesting and scientifically sound.

However, the following major points should be addressed by the Authors before it can be reconsidered for publication:

  1. Although the paper is clearly focused on mitochondria-derived oxidative stress, for the sake of clarity, Authors should adequately discuss whether other sources of ROS have been possibly implicated in ALS pathogenesis.
  2. An aspect that, in my opinion, could improve the paper is the reference to possible evidence in animal models of ALS which might support the findings obtained on the used cell line.
  3. Authors should further and deeper elaborate the possible impact of their findings on the clinical settings and the translational approach that may derive from their findings.

As minor point, the paper should be revised for typo errors corrections (for ex. line 47, a space is missing between SOD1 and have).

Author Response

Reviewer #4 (Remarks to the Author):

In this paper, Authors aimed to assess the ability of oxidative stress observed in ALS to induce post-translational modifications (PTMs) in VDAC1, the main protein of the outer mitochondrial membrane. The experimental procedure was performed by using mitochondria of a NSC34cell line expressing human SOD1G93A, a cellular model of ALS and, based on the obtained results, Authors claimed the presence of significant changes to the structure of the VDAC1 channel and to the bioenergetics metabolism of ALS motor neurons.

I found this paper interesting and scientifically sound.

However, the following major points should be addressed by the Authors before it can be reconsidered for publication:

  1. Although the paper is clearly focused on mitochondria-derived oxidative stress, for the sake of clarity, Authors should adequately discuss whether other sources of ROS have been possibly implicated in ALS pathogenesis.

We thank the referee for raising this point. Actually, dysfunctional mitochondria represent the major source of disproportionate ROS/RNS production. Oxidative stress caused by mitochondrial ROS/RNS imbalance is a key hallmark of ALS that in turn can worsen mitochondrial dysfunction in a sort of vicious circle. Following your advice, we added in the “Introduction” some considerations about non-mitochondrial ROS/RNS.

  1. An aspect that, in my opinion, could improve the paper is the reference to possible evidence in animal models of ALS which might support the findings obtained on the used cell line.

There are no evidence in literature about protein deamidation in animals, but only in few patients (see discussion). This is possibly due to the lack of appropriate searches for this rare PTM.

  1. Authors should further and deeper elaborate the possible impact of their findings on the clinical settings and the translational approach that may derive from their findings.

We have outlined in discussion some hypothetical role of deamidation as a signal of mitochondrial quality control.

- As minor point, the paper should be revised for typo errors corrections (for ex. line 47, a space is missing between SOD1 and have).

We thank the reviewer, we checked the msc for typos.

Round 2

Reviewer 2 Report

This referee does not agree with the experimental model the authors decided to use. Statements like ..."it is not so easy to have primary ALS model neurons available. In fact, to represent a valid model of the disease, the transgenic ALS mouse should be available; alternatively, primary motor neurons cultures taken from a wt mouse and then transfected with a construct expressing SOD1 G93A should be used. In this last case, however, the transfection efficiency of the primary neurons is usually very low" are not acceptable.

Reviewer 3 Report

Authors have addressed my comments.

Reviewer 4 Report

I am fine with the answers provided by the Authors to my comments and with the revisions they performed accordingly.